# The Impact of Practicing Open- vs. Closed-Skill Sports on Executive Functions—A Meta-Analytic and Systematic Review with a Focus on Characteristics of Sports

**DOI:** 10.3390/brainsci12081071

**Published:** 2022-08-12

**Authors:** Florian Heilmann, Henrietta Weinberg, Rainer Wollny

**Affiliations:** 1Movement Science Lab, Institute of Sport Science, Martin-Luther University Halle-Wittenberg, 06120 Halle (Saale), Germany; 2Movement and Sport Psychology, Institute for Sport Science, Friedrich-Schiller-University Jena, 07749 Jena, Germany

**Keywords:** executive functions, cognitive functions, sport modes, cognitive skill transfer

## Abstract

Exercise modes can be categorized based on the skills required (open vs. closed skills), which implicates various demands on cognitive skills, especially executive functions (EFs). Thus, their practice may have varying effects on EFs. There is a lack of detailed analysis of cognitive requirements and suitable classification of sports. It is hypothesized that the amount and type of cognitive requirements of sports lead to small effect sizes when comparing open-skill exercising (OSE) and closed-skill exercising (CSE) athletes. The current meta-analysis evaluates the variances in EFs skills caused by particular sport modes. Four research databases (Web of Science, PubMed, ScienceDirect, PsychINFO) were searched for cross-sectional studies in which the authors compare the performance in EF tasks of OSE and CSE athletes. Risk of bias assessment was conducted using funnel plots and two reviewer selection process (overall and subgroup analysis; low risk of publication and selection bias). A total of 19 studies were included, revealing an overall effect size of Hedge’s g = 0.174 (*p* = 0.157), favoring OSE for the development of EFs. The subgroup analysis revealed the effects for the subdomains of EFs (cognitive flexibility: Hedge’s g = 0.210 > inhibitory control: Hedge’s g = 0.191 > working memory: Hedge’s g = 0.138; *p* > 0.05), which could be characterized as low to moderate. The hypothesis that studies with the smallest effect sizes compare sport modes with similar cognitive demands was rejected. The paper discusses the differentiation of sports into OSE and CSE and presents new approaches for their categorization.

## 1. Introduction

Executive functions (EFs), including cognitive skills such as working memory (WM), inhibitory control (IC), cognitive flexibility (CF), planning, reasoning, and problem-solving, enable humans, especially athletes (for definition of “athlete” see [1]), to display goal-directed behaviors, adapt to novel situations, and manage social interactions [2]. Research in sports science reveals that the expression of EFs can benefit from physical activities (for reviews, see [3,4]) and the exercise related to sport [5] or could be impacted by performing certain types of sports [6]. Numerous studies indicate that athletes have better EF performance than non-athletes (for review, see [7]). Recent studies have evaluated the differences in EF between athletes with high expertise in open- or closed-skill exercise (OSE vs. CSE) and classified open-skill exercise as superior in terms of EF performance [6,8]. OSE are in a dynamic environment, where conditions could change at any time (e.g., nature or team sports), and CSE are characterized as static with defined and known conditions (e.g., swimming or running).

A few studies extended the research question from a simple comparison between athletes’ and non-athletes’ expression of EFs to implementing a tangible sports performance (soccer performance: [9]; ice hockey performance: [10]). The current meta-analytic review investigated the effect of practicing open- or closed-skill exercises on the expression of EFs. Furthermore, the characteristics of closed- and open-skill exercises are investigated, and subgroup and moderator analysis are performed to determine if the effects are more or less critical for different subdomains of EFs (working memory, inhibition, cognitive flexibility) and if the age of samples affects the outcome.

### 1.1. Executive Functions and Sports Experience

Previous research revealed that practicing a sport could support the expression of EFs. The findings of studies relating to the sports of soccer [11,12], tennis [13], basketball [14,15,16], and volleyball [17,18] support this hypothesis. The studies examined the difference between athletes’ and non-athletes’ performance of EFs and reported that athletes show superior EFs compared to non-athletes. The hypothesis could be expanded, because numerous studies indicated differences in EFs based on the particular type of sport that they chose to study (open vs. closed skill).

### 1.2. Effects of Open- and Closed-Skill Sport Practice

Sports science, especially movement science, differentiates sporting movements based on their special environmental and task requirements and the resulting degrees of variability in movement execution on a horizontal continuum within the extremities of closed and open skills [8,19,20].

Gymnastics, track and field, swimming, and shooting could be characterized as closed-skill sports [20,21,22,23]. In these sports, a particular skill, such as the Biellmann pirouette in figure-skating or the triple somersault in water jumping, is often the goal and purpose of the movement itself. In Category 1 of closed-skill sports, the form of movement is fairly fixed for the specific type of sport, and the environmental and task requirements are primarily constant during the execution of the movement (e.g., gymnastics; [20]). In Category 2, which comprises the continuum of closed to open skills, the environmental conditions are already known (e.g., athletics disciplines; [19]) and so could be implemented in the pre-existing program of movement. A similar continuum model without the specified categories was introduced by Honeybourne [19].

Tennis, soccer, alpine skiing, and surfing could be defined as open-skill sports. These skills often serve to achieve a goal and purpose that are independent of their external form (e.g., skiing on a mogul slope). Open-skill sports are characterized by a wide range of variations and a low level of dependency on certain specific movement sequences (e.g., combat, team, and nature sports such as windsurfing and alpine skiing). In Category 3 of open-skill sports, athletes can foresee situational conditions to a limited extent only (e.g., in nature sports such as surfing and skiing). In Category 4, athletes cannot at all predict the diverse environment (e.g., combat and team sports) and so have to react very quickly and dynamically to constantly changing movement requirements.

The environmental conditions or requirements of movements could impact EFs of practitioners. For example, Furley and Memmert [14] have shown that expert basketball players can resist competing stimuli and focus on the task in a better manner than novices. With this skill, they can make better tactical decisions than amateurs. Closed-skill sports are performed in conditions that remain primarily constant and are known or irrelevant for the course of movement. Thus, the athletes do not have to adapt to changing environmental conditions as often as in open-skill sports. For these reasons, CSEs are considered to have lower demands on cognitive skills. OSE could lead to a better performance in EF tasks because the particular sport modes require higher cognitive demands, especially with regard to EFs. For example, if a soccer player plans a pass to a teammate and an opponent covers the passing line, the player must inhibit his initially planned movement (inhibition) [9].

The underlying theory in the current study is the broad skill transfer hypothesis [24]. This approach assumes that the cognitive skills achieved while training for or practicing a particular sport will also be transferred to untrained cognitive performance or tasks. Thus, in the example of OSE, a training to switch between different strategies in tennis (baseline vs. serve and volley) would lead to higher performances in CF in laboratory.

Many studies show positive correlations between practicing open-skill sports and EF performance [8,25,26]. Studies that examined EFs in OSE report that OSE are superior to CSE in the context of development of EFs (e.g., [26,27]).

According to Yongtawee [28] and Krenn [25], when classifying sports as interceptive (boxing), strategic (soccer), or static (gymnastics) based on the interaction between teammates and opponents and the dynamic environment, interceptive and strategic sports athletes tend to show better processing speed and cognitive skills [29] when compared to those who engage in static sports. Another classification of sport mode suggested by Ballester [30] is externally paced (e.g., baseball) and self-paced (e.g., swimming). The definition of these categories is similar to that of OSE and CSE.

Overall, the classifications of CSE and OSE are not exhaustive with respect to the impact on cognitive functions, but they could lead to a general differentiation. However, for example, even the tripartite classification by Yongtawee et al. [28] could not differentiate between static sports with high and low cognitive requirements (e.g., gymnastics vs. athletics).

There is an ongoing debate about the interaction between requirements of various sport types and EFs and whether only OSE could support the development of EFs. Furthermore, there is a lack of knowledge concerning the relationship between performance in closed-skill sports and EFs and concerning possible mechanisms of promoting EFs.

Recent studies by Zhu et al. [31] and Gu et al. [32] investigated the effect of OSE versus CSE on various cognitive functions. They described the type of sport with OSE and CSE but used a limited subdivision that does not consider minor differences in cognitive demand and specific requirements of sport modes. The subgroup analysis focused only on the type of cognitive function or on the age of samples. The authors studied a broad range of cognitive functions: inhibitory control, cognitive flexibility, visuospatial attention, visuospatial working memory, processing speed, problem-solving, vigilance, and decision making [30]. Their studies did not clarify if CSE with high cognitive demands could lead to more or less the same improvements in EFs as OSE.

Therefore, the current systematic review aims to evaluate the characteristics of sports evaluated in previous observational studies (using the continuum model described above) to identify the processes that can lead to an improvement in EFs. First, the differences between OSE and CSE’s effects on EFs were retrieved from the latest research findings (updated literature search). For this purpose, a literature search was conducted in relevant databases. Then, the identified records were synthesized in a quantitative meta-analysis.

Based on the cognitive skill transfer hypothesis [24], we hypothesized that there could be reported differences between the EFs of OSE and CSE athletes. The main difference that we considered was the focus on the examined sport modes in the included studies. Using this approach, we tried to find explanations for differences in the expression of EFs. We hypothesized that the effect sizes of group differences in studies on sport modes within categories 2 and 3 (e.g., athletics vs. canoe slalom) are smaller when compared to studies on sports within categories 1 and 4 (e.g., swimming vs. basketball). The differences should be smaller, because the sports have similar cognitive requirements and train homogenous cognitive skills. This hypothesis suggests that the cognitive requirements of the sport should be considered when interpreting the effects of CSE vs. OSE. The differentiation between categories is conducted on a descriptive level in this study. In addition, the sample’s age and study quality should be regarded as moderators while performing the meta-regression analysis.

## 2. Materials and Methods

### 2.1. Literature Search Strategy

The literature search was conducted using systematic review and meta-analysis, as recommended by the PRISMA statement for searching and reporting studies [33]. The search was performed using a combination of the following terms: (a) “executive function” (title search; MeSH Terms); (b) “open-” or “closed-skill”, “self-“ or “externally paced”, “static”, “interceptive”, “strategic sport” or “exercise mode”; and (c) “sports” (title search; MeSH Terms). Cross-sectional studies that reported relevant findings about the executive functions of OSE vs. CSE athletes were identified in the search of databases Web of Science, PubMed, ScienceDirect, and PsychINFO. MeSH term search was conducted for PubMed database. The research began with literature search at the beginning of October 2021 and ended with statistical analysis at the end of October 2021. Additionally, reference lists of included articles were created for further studies after confirming their eligibility.

### 2.2. Selection Criteria

Full-text articles written in English language, published between 1 January 2000 and 1 October 2021, were considered eligible for inclusion in the qualitative and quantitative analysis. Records were selected if they (a) examined athletes of open- and closed-skill sports (min. of one sample group per category), (b) stated a level of performance for the examined athletes, (c) stated the age of the participants, (d) measured at least one of the three core EFs (working memory, inhibitory control, cognitive flexibility), (e) used a cross-sectional design for analysis, (f) reported the differences of open- and closed-sports athletes using statistical analysis, and (g) presented the results in terms of mean, SD, and sample size (by considering open and closed skills as independent variables). Studies that (a) had medical questions, (b) included disabled or impaired subjects in the study, or (c) only used self-report or other psychometric measurements (questionnaires or ratings) for characterization of EFs were excluded from the analysis.

### 2.3. Qualitative and Quantitative Analysis

The qualitative analysis included a comparison of studies based on outcome measures and test statistics to identify the eligible studies. In addition, study designs and samples were compared and analyzed in the context of the research question. Specifically, the exercise mode and type of sport were included in the discussion. The conducted tests were taken into consideration for the qualitative analysis of the systematic review.

The quantitative synthesis analyzed studies that reported effect size, test statistics, or result tables using means, SD, or SE, along with sample sizes for groups. Quantitative synthesis was conducted based on Hedge’s g to correct for small sample sizes and Bessel correction for different sample sizes (n − 1; Formula (1)).
(1)$$g=\frac{{\bar{x}}_{1}-{\bar{x}}_{2}}{\sqrt{\frac{\left(n_{1}-1\right)\times s_{1}^{2}+\left(n_{2}-1\right)\times s_{2}^{2}}{n_{1}+n_{2}-2}}}$$

Subtasks in the studies with respect to different parameters were summarized based on overall effect size. Different categories of EFs (inhibitory control, working memory, and cognitive flexibility) were considered as subgroups in the meta-analysis. If the authors reported various measures for one subgroup, the combined effect for the EF was calculated. Overall effect sizes and effect sizes of single-test studies were applied to random-effect model meta-analysis (*k* = 19). Omnibus test statistic *Q* was calculated to determine the variability in the distribution of effect size estimates, as performed by [34]. Heterogeneity was indicated by *I*^2^ and tested for the overall analysis. For every subgroup, the respective EF was calculated. Hedge’s g was interpreted based on Cohen’s guideline [35], which suggests that 0.2 be considered a small effect, 0.5 be considered as a moderate effect, and 0.8 be considered a large effect. Thus, an *I*^2^ of 25% was characterized as low, 50% as moderate, and values greater than 75% as highly heterogeneous [36]. The significance of the *p*-value was set at *p* < 0.05.

Moderator analysis was conducted using a mixed-effect model by considering Tau^2^ and Hedge’s g as estimators of overall effect (age and study quality) and subgroup analysis (age). Study quality was not considered as a moderator of the subgroup analysis, as it has no relevance in terms of content. In addition, calendar age and study quality were established as continuous moderator variables (CI interval level: 95%).

Overall effect sizes (Hedge’s g) and test statistics of meta-analysis were calculated in Rstudio (Version 1.4.1103, Rstudio PBC, Boston, MA, USA), using the “esc” package. Meta-regression was performed using the Jamovi software (Version 2.2.2, jamovi Project, https://www.jamovi.org, accessed 10 October 2021) and package “meta”.

### 2.4. Risk of Bias Assessment

Two independent reviewers (F.H. and H.W.) were designated to conduct the literature search to avoid selection bias. They screened the titles, abstracts, and full texts of identified records and rated the quality of studies using the NIH study quality assessment tool (NIH, 2021, https://www.nhlbi.nih.gov/health-topics/study-quality-assessment-tools, accessed 10 October 2021). In the event of discrepancies, a third reviewer was brought in to rate the particular study. Publication bias was studied using descriptive analysis of a funnel plot. Asymmetry or incompleteness of the funnel plot was interpreted as an indication of publication bias [34].

## 3. Results

### 3.1. Study Selection

A flow diagram of the selection process is presented in Figure 1. The search was conducted in four electronic databases, namely Web of Science, PubMed, ScienceDirect, and PsychINFO. Studies of previous reviews [31,32] on the differences in cognitive functions (e.g., EFs) achieved through exercise were included in the qualitative and quantitative analysis (first strategy). After the screening of articles on previous studies, eleven studies of the previous reviews were identified as eligible. Six out of seventeen studies were excluded, as the authors of the reviews had included intervention studies in their analysis. These studies were not in the focus of the current study.

At the end of the main database search, 8812 potential articles were identified for the current study (second strategy). After the removal of duplicates, 5920 articles remained. After title and abstract screening, 26 titles remained for full-text search. Eighteen records were excluded because of the following reasons: (1) they had already been referenced in previous reviews (n = 11; [8,22,26,30,37,38,39,40,41,42,43]), (2) data were not available even after requesting it from the corresponding author (n = 3; [6,44,45]), (3) EF measure was included only for higher EFs (n = 1; [46]), (4) the EFs were categorized in an inappropriate form (n = 1; [25]), or (5) the identified record was an intervention or treatment study (n = 2; [47,48]), or no differences for OSE vs. CSE were calculated (n = 1; [49]). Three additional records were identified by cross-checking the references in the studies during full-text screening (third strategy).

After an independent evaluation of full texts based on the predefined inclusion criteria, seven new studies were included in the review (two reviewers agreed on the criteria). Thus, a total of 19 studies were selected for qualitative and quantitative analysis, systematic review, and meta-analysis. These included articles are listed in Appendix A. The quality of included articles was rated at 7.78 (Mdn: 8 [6,7,8,9]; SD: 0.69), with 100% agreement among all reviewers.

### 3.2. Results of Qualitative Synthesis

The 19 identified articles were published between 2008 and 2021. Four of the nineteen studies (21.05%) evaluated the effect of sports on EFs for only one gender [26,50,51,52].

One of the nineteen studies examined children (5.26%), one studied preadolescents (5.26%), one studied adolescents and adults (5.26%), ten studied the EFs of only adult athletes (47.37%), and seven studied older adults (36.84%). The characterisation of included studies is displayed in Appendix A.

Inhibitory control (IC) was examined in twelve studies (63.15%), working memory (WM) was assessed in eight studies (42.11%), and cognitive flexibility (CF) was evaluated in six studies (40%). The authors used various tasks to measure EFs. The most commonly used tasks were Stroop task (for IC), Eriksen flanker task (for IC), visuospatial memory task (for WM), n-back task (for WM), and trail-making task (for CF).

The sport modes that were compared in the studies are displayed in Table 1 The most frequently evaluated sports for OSE were tennis (18.18%), table tennis (18.18%), and badminton (13.64%), and those for CSE were swimming (32.61%), running (26.09%; no trail running), and athletics (15.22%). Fourteen of the included studies used a control group (73.68%), while five studies did not use any control group in their study (26.32%, see Appendix A). One study evaluated differences after the completion of hours of training per week [30] and an International Physical Activity Questionnaire (IPAQ; examines exercise behavior and time spent sitting) and reported superior performance by the CSE group [53]. In three articles, a comparison of the level of performance or experience between the two groups was not entirely reported [27,50,52]. Fourteen studies reported that the OSE and CSE groups did not show any difference in their level of performance or sports experience.

Ten of the included studies (52.63%) concurred with the postulation of previous studies that OSE athletes show superior EFs when compared to CSE athletes. The findings of five studies (26.32%) were inconsistent. Four studies (21.05%) could show superiority of neither OSE nor CSE athletes with regards to better EFs, etc.

### 3.3. Results of Quantitative Synthesis

#### 3.3.1. Overall Analysis

The random-effect model (*k* = 19), which included all studies with combined effect sizes, showed an estimated overall effect size of *g* = 0.174 (95% CI [−0.067, 0.415]; SE = 0.119; Figure 2. The estimated effect was not significant (*p* = 0.157). The analysis of heterogeneity was also not significant (*Q*(19) = 2.257; *p* = 1.000; Tau^2^ = 0.000; *I*^2^ = 0%; *H*^2^ = 1.000). The lowest effect sizes were negative (*g* = −0.15 to −0.03, favoring CSE). The highest effect sizes of the model reached *g* = 0.47 to 0.65. The results of the overall meta-analysis are displayed in the forest plot in Figure 2.

Calendar age was not a moderator in the analysis (mixed model effect: −0.002; *p* = 0.686; 95% CI [−0.012, 0.008]; SE = 0.005). Study quality was also not a significant moderator of the meta-analysis (mixed model effect: −0.037; *p* = 0.818; 95% CI [−0.356, 0.282]; SE = 0.163).

#### 3.3.2. Subgroup Analysis

In the subgroup analysis, the effect sizes of the random-effects model for the three EFs (inhibitory control, working memory, cognitive flexibility) were calculated independently. The results of this analysis are shown in the forest plot in Figure 3.

The random-effect model for the studies evaluating the effect of OSE versus CSE participation on inhibitory control (*k* = 11) revealed an overall effect size of *g* = 0.191 (*p* = 0.094; 95% CI [−0.117, 0.500]; SE = 0.157). Heterogeneity analysis showed no significant results (*Q*(10) = 0.773; *p* = 1.000; Tau^2^ = 0.000; *I*^2^ = 0%; *H*^2^ = 1.000). Calendar age did not affect subgroup analysis for inhibitory control (mixed model effect: −0.003; SE = 0.006; *p* = 0.679).

The effect of subgroup analysis of working memory revealed an overall effect size of 0.138 (*p* = 0.474; 95% CI [−0.240, 0.516]; SE = 0.193). Analysis of heterogeneity of subgroup analysis returned a small value (*Q*(6) = 1.793; *p* = 0.938; Tau^2^ = 0.000; *I*^2^ = 0%; *H*^2^ = 1.000). Calendar age did not affect subgroup analysis for working memory (mixed model effect: −0.001; SE = 0.008; *p* = 0.895).

The effect size estimate of the cognitive flexibility of OSE and CSE athletes was not significant (*p* = 0.276; *g* = 0.210 *p* = 0.276; 95% CI [−0.168, 0.587]; SE = 0.193). Analysis of heterogeneity of subgroup analysis returned a small value (*Q*(7) = 0.519; *p* = 0.999; Tau^2^ = 0.000; *I*^2^ = 0%; *H*^2^ = 1.000). The moderating effect of calendar age on overall effect size with respect to cognitive flexibility was −0.005 (SE = 0.008; *p* = 0.505).

The funnel plot analysis (Figure 4) did not indicate publication bias for overall and subgroup effect size estimates. An asymmetry in data points could not be determined with respect to the overall effect. No effect size or SE exceeded the 95% confidence interval. Furthermore, no specific gap in the variation in studies could be identified.

## 4. Discussion

The current systematic review and meta-analysis aimed to assess the overall effect of the impact of OSE and CSE participation (categories 1–4) on EF performance. To the best of our knowledge, this is the first qualitative and quantitative synthesis of studies that analyses the effect of sport modes on EFs only. Furthermore, previous reviews did not mainly focus on the characteristics of examined sport types. The subgroup analysis is updated with the latest research on the effects of different sport modes on the expression of EFs (inhibitory control, working memory, cognitive flexibility). An important question regarding the sport mode is which classification of sport types is applied to describe differences in cognitive functions. In previous studies, bivariate comparison between open-skill and closed-skill sports was dominant (with redundant or similar bivariate division into external and self-paced sports). Yongtawee et al. [28] and Krenn et al. [25] suggested a tripartite classification. We applied our own characterization for the current study to describe the variance between sport modes more precisely using four categories in the qualitative analysis.

### 4.1. Discussion of Qualitative Synthesis

As far as qualitative synthesis is concerned, relatively few studies had samples with a low average calendar age (children, adolescents). However, this is important, because it is at this age (3–18 years) that the formation of EFs is highly relevant and so should be recorded and researched with reference to longitudinal studies [45].

The studied sport modes did not vary widely. The most frequently studied sports had similar characteristics (i.e., tennis, table tennis, badminton). This applies to both OSE and CSE. The studies lacked detailed description and sport-mode-specific interpretation of results. The qualitative analysis revealed that OSE was evidently superior to CSE in terms of EFs, as proved by 10 out of 19 studies, which showed positive effects for OSE.

The frequently examined OSEs such as tennis, table tennis, and badminton were classified into category 4, which entails high cognitive requirements, variable environmental conditions, and erratic movements. In contrast, the commonly examined CSEs, such as swimming and running, which do not require high cognitive functions, were classified into category 1. Due to the different cognitive requirements in these categories, the effect of CSE vs. OSE could be overestimated when comparing category 1 with category 4. However, the effect should be much higher than when comparing categories 2 and 3.

### 4.2. Discussion of Quantitative Synthesis

#### 4.2.1. Overall Analysis

The broad skill transfer hypothesis, which is the suggested theory to explain the phenomena occurring in this context, argues that training in sports or cognitive tasks may increase performance in related but untrained tasks [24,54]. Recent studies confirm this theory [11]. For example, [8] postulates that OSE athletes are superior in EFs to CSE athletes because they are trained better in the relevant cognitive performances. The overall effect size calculated to determine the difference in EF measures between OSE and CSE athletes was indicated as low (*g* = 0.174; *p* = 0.157) and not significant. Based on the proven homogeneity of the studies, it can be concluded that OSE athletes have moderate advantages over CSE athletes in their executive functions. This finding also coincides with the results of the studies by Gu et al. [32] and Zhu et al. [31]. The analysis of further studies has expanded the evidence. The overall effect size could be confounded by the selection of sport modes and the design of the study. It stands to reason that studies that analyze differences in EFs in sports such as badminton and swimming can show high effect sizes or mean differences.

#### 4.2.2. Characterization of Examined Sport Modes

It is particularly striking that the five studies with the smallest effect sizes [22,38,39,40,41] show the difference between OSE sports such as tennis, table tennis, and badminton (category 4) and CSE sports such as swimming, running, and triathlon (category 1). The OSE sports considered in this case were racket games, which were characterized by one or two opponents at the maximum. The studies with the highest effect sizes [27,43,50,52,53] compared various OSE sports such as basketball, canoe slalom, handball, Olympic sailing, and baseball with two or more CSE sports such as archery, cross-country skiing, shooting, speed-skating, and weightlifting.

Furthermore, while some activities could be characterized obviously more as OSE and others more as CSE, this distinction is not always made clear in the studies. For example, why are sailing and canoe slalom categorized as OSE (i.e., using a boat to navigate a route while evading obstacles or other competitors), while cross-country skiing is characterized as CSE (i.e., using skis to navigate a route while racing other competitors)?

Even though these studies have sample sizes exceeding the value of 75 (n > 75), the mixing of many sports into groups could lead to an opaque attribution of the effects [24]. The hypothesis that effect sizes of studies comparing athletes of sports categories 2 and 3 are smaller than those comparing athletes in sports categories 1 and 4 had to be rejected. Differences in effect size could not be explained by the selected sport mode. Therefore, at this point, further research is needed to arrive at a strong conclusion. Furthermore, in order to be able to make an unequivocal comparison between the studies, the requirements in the specific EF task must be considered. Currently, this is only done through the division between the different cognitive abilities.

#### 4.2.3. Subgroup Analysis

One area in which previous studies were lacking is that they did not calculate the effect of sports on working memory. The subfunction or subgroup analysis studied visuospatial attention and processing speed instead of working memory [31]. The subgroup analysis of the current study shows differences in the effect size between the three EFs (CF: *g* = 0.210 > IC: *g* = 0.191 > WM: *g* = 0.138). The different requirements may explain this order of effect of OSE for CF, IC, and WM. It could be speculated that OSE athletes often have to switch between strategies or tasks [39] and inhibit irrelevant information [11].

## 5. Limitations

Although no potential selection or publication bias could be reported, a few limitations of the current meta-analytic review must be reported. First, only the cognitive demands were considered when examining the effects of OSE and CSE on EFs, not the physical fitness (e.g., aerobic or muscular fitness), level of performance or competition, or levels of endocrine hormone status reported in the studies. Due to the heterogeneous types of measurement and the description of the mentioned factors, they cannot be included in the calculation. Secondly, the type of practice could not be considered in this study. Of course, in OSE, athletes sometimes have to train skills in a closed-skill setting, especially in the first years of practice, but this fact could not be examined in the current analysis. Thirdly, longitudinal studies have not been included in this review and meta-analysis. This could impact the results of the investigation. The effects measured in intervention or longitudinally designed studies are affected by the specific character of the intervention. Considering the particular type of intervention in the calculation and the still-low availability of such studies represent the issues with considering this factor.

## 6. Conclusions

This current meta-analysis could not validate the hypothesis that sport modes examined in the included studies representing low to moderate effect sizes have similar cognitive demands. However, the commonly examined sport modes differ with respect to demands of skills. The design of the current study does not lend itself to the investigation of this research question to the fullest extent. Given this drawback, it is quite probable that the overall effect was overestimated. In this context, [25] suggested in their article that future research must study different cognitive demands in OSE and CSE. The results of the current meta-analysis suggest a completely new approach to this issue. Future research has to classify the included sport modes based on their cognitive demands to prove the broad skill transfer hypothesis [24] in this context. Simple differentiation between OSE and CSE cannot pave the way for further inferences in this field. The presented classification is suggested for future research. A conceivable study design could examine EFs in two sport modes of OSE and CSE, with differing cognitive demands or skills. In this case, OSE with low demands should not differ from CSE with high cognitive requirements (i.e., sports of mentioned categories 2 and 3: athletics vs. alpine skiing). Considering the different effect sizes for the impact of requirements in sport on IC, WM, and CF, it is helpful to implement tasks for these three parts in future research. This approach can determine to what extent these differences can be empirically proven. Furthermore, research has to deal with different coaching methods. It would be interesting to compare EFs of athletes practicing in a closed- vs. open-skill development.

## Figures and Tables

**Figure 1 brainsci-12-01071-f001:**
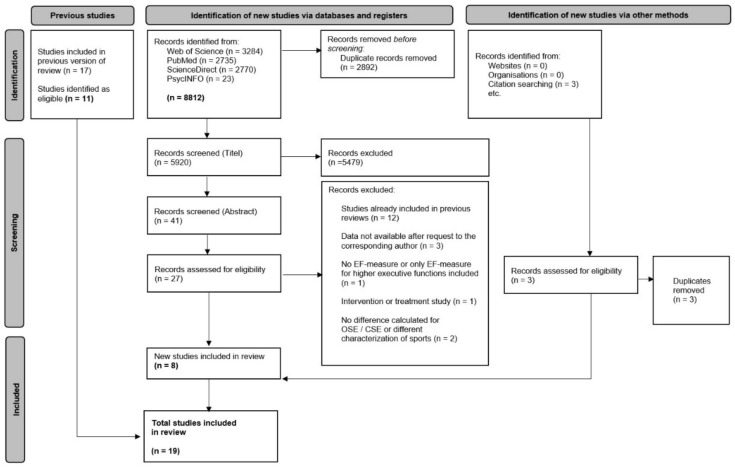
PRISMA Flowchart of the study selection process.

**Figure 2 brainsci-12-01071-f002:**
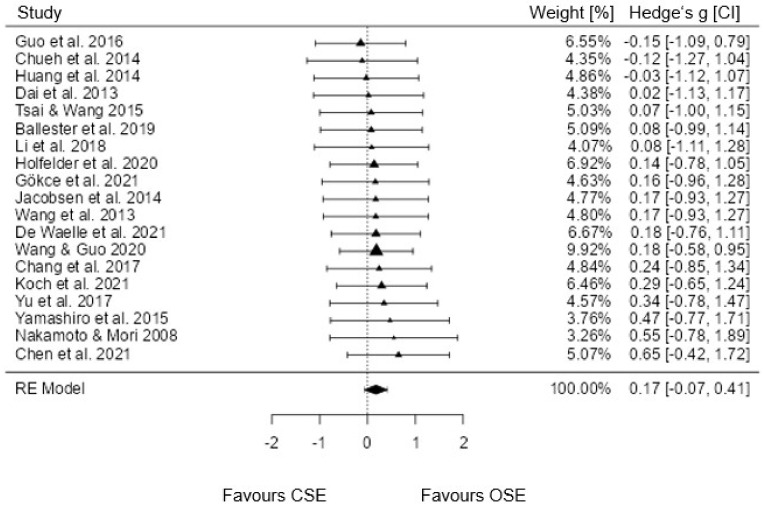
Forest plot for the efficacy of OSE compared to CSE (ordered according to the specified effect size); RE = random effect model [8,25,26,27,30,37,38,39,40,42,43,50,51,52,53,54,55,56,57].

**Figure 3 brainsci-12-01071-f003:**
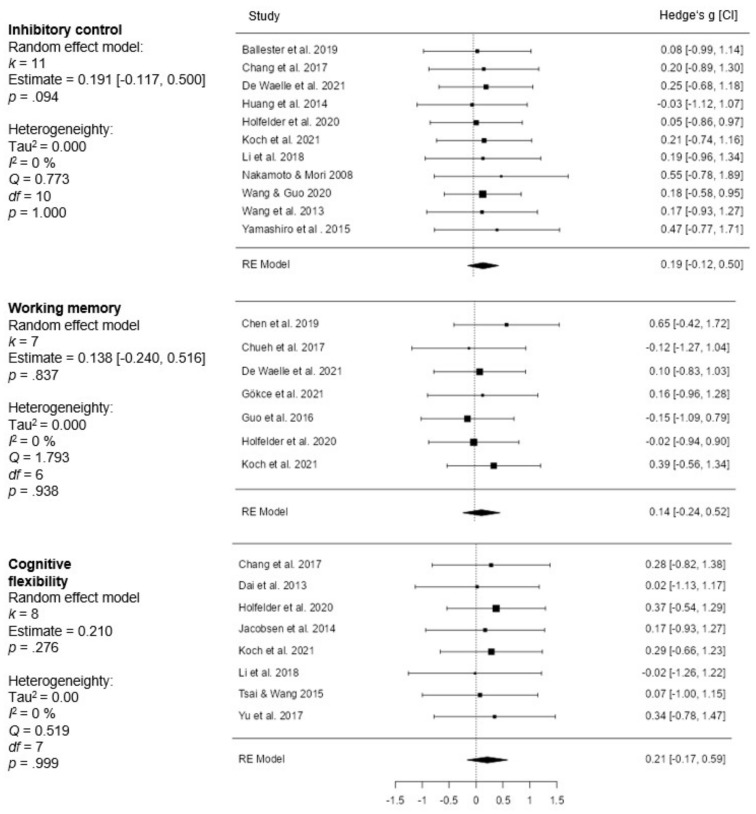
Forest plots of subgroup analysis (OSE vs. CSE); RE = random effect model [8,25,26,27,30,37,38,39,40,42,43,50,51,52,53,54,55,56,57].

**Figure 4 brainsci-12-01071-f004:**
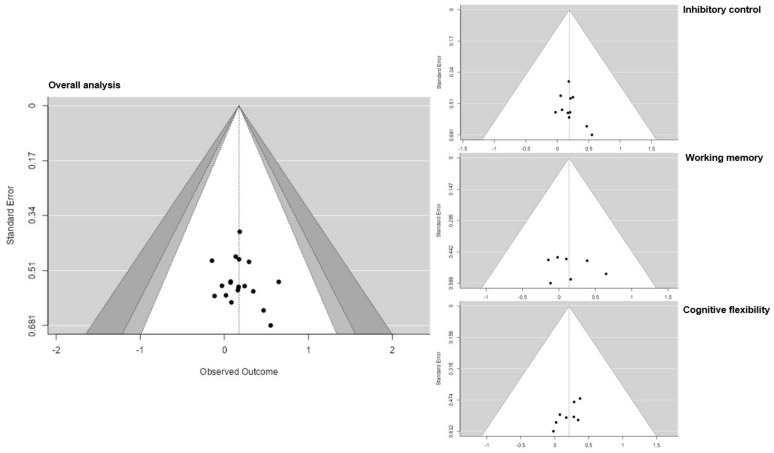
Funnel plots for overall analysis, inhibition, working memory and cognitive flexibility.

**Table 1 brainsci-12-01071-t001:** Frequency and percentages of OSE and CSE examined and compared in the included studies.

OSE	Classification	Frequency *N* = 44	Percentage [%]	CSE	Classification	Frequency *N* = 46	Percentage [%]
Tennis	4	8	18.18	Swimming	1	15	32.61
Table Tennis	4	8	18.18	Running	1	12	26.09
Badminton	4	6	13.64	Athletics	2	7	15.22
Basketball	4	5	11.36	Triathlon	1	3	6.52
Volleyball/Beach volleyball	4	2	4.55	Cycling	1	3	6.52
Soccer	4	3	6.82	Gymnastics	1	1	2.17
Handball	4	2	4.55	Archery	1	1	2.17
Sailing	3	2	4.55	Shooting	1	1	2.17
American Football	4	1	2.27	Brisk walking	1	1	2.17
Wushu	4	1	2.27	Cross-country skiing	2	1	2.17
Martial Arts	4	1	2.27	Track-bike	1	1	2.17
Fencing	4	1	2.27				
Korfball	4	1	2.27				
Hockey	4	1	2.27				
Canoe slalom	3	1	2.27				
Baseball	4	1	2.27				

## Data Availability

The data presented in this study are available on request from the corresponding author.

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
