# Peer review of "The Impact of Practicing Open- vs. Closed-Skill Sports on Executive Functions—A Meta-Analytic and Systematic Review with a Focus on Characteristics of Sports"

_brainsci, 2022, doi:10.3390/brainsci12081071_

Round 1

Reviewer 1 Report

Overall, the study aims to address interesting topic and analysis is well done. However, there are some points that need to be improved. First, introduction in general is unclear. It should be more clear and specific. Second, definition of athlete is questionable. Third, I have a concern about the high heterogeneity and relatively low quality of studies included in this review study. Specific comments are below.

Abstract

1.      Lines 14-15: ‘similarity of cognitive requirements’ is not clear. Please clarify.

2.      Line 18: Cross-sectional studies are not able to investigate ‘effects’. Authors need to clarify whether they are referring to moderating effects or association.

3.      Line 22: Rather than saying ‘various EFs’, different subdomains of EFs make more sense to me.

Introduction

1.      Line 38: Need a brief explanation regarding OSE and CSE.

2.      Lines 49-50: Which kind of EFs? Need to clarify specific subdomains of EFs.

3.      Lines 58-60: This statement needs reference(s).

4.      Lines 70-78: This paragraph needs reference(s).

5.      Lines 61-78: To me, the mentioning of ‘Category’ is pretty abrupt, meaning that they were presented without background information. It would be great if authors can add some more information about this.

6.      Lines 79-80: Since this study is focusing on executive function, executive function can appear from the first, not starting with cognitive function. There are so many subdomains under cognitive function. Starting with executive function will make the readers more focused.

7.      Lines 87-91: To my understanding, OSE leads to higher cognitive function because it induces greater cognitive demand? Please clarify this in this paragraph.

8.      Lines 131-132: Again, need to clarify categories 1-4.

9.      In general, the introduction is quite long and there are too many paragraphs. Authors need to make it more concise with clear background information, gaps in knowledge, aims and hypothesis.

Methods

1.      Calendar year was used to examine the moderating effects.  This does make sense given the variability in the age range across the studies. However, considering that this study is focused on athletes, their practicing year is as important as their calendar year?

2.      Lines 250-251: Can training hours/week and IPAQ be used as to determine ‘athletes’? The definition of ‘athlete’ should include both high duration and intensity of training as well as participating in regional/national/international competitions. This definition tool questions me if the studies the authors selected can be analyzed under the same category of athletes.

Discussion

1.      Line 356: I could follow the previous sentences regarding category 1-4, but I could not follow the last sentence. What does it mean by ‘… overestimated in a narrow sense’?

2.      Lines 374-381: For the OSE sports including tennis, table tennis, and badminton, they are more 1 to 1 sports wile basketball and handball are more team sports. Do authors think the individual vs team sports affect the present analysis?

3.      Line 397-398: The authors just stated that working memory was not studied in an independent manner? Do authors think working memory should be studied separately? If so, what is the rationale?

4.       

Author Response

Comments to Reviewer 1

[General Comments] First, introduction in general is unclear. It should be more clear and specific.

Second, definition of athlete is questionable.

Third, I have a concern about the high heterogeneity and relatively low quality of studies included in this review study.

[Comment 1] Lines 14-15: 'similarity of cognitive requirements' is not clear. Please clarify.

Response: Thank you very much for this specific comment. We changed the word "similarity "to "the amount and type of cognitive requirements "because we found the wording confusing. Furthermore, we refer the requirements to the exercising athletes and the different sports.

[Comment 2] Line 18: Cross-sectional studies are not able to investigate 'effects'. Authors need to clarify whether they are referring to moderating effects or association.

Response: You are right. Furthermore, the studies themselves could not compare effects. Therefore, we revised the passage as follows:

"The current meta-analysis evaluates the variances in cognitive skills caused by particular sport modes. Four research databases (Web of Science, PubMed, ScienceDirect, PsychINFO) were searched for cross-sectional studies in which the authors compare the performance in EF tasks of OSE and CSE athletes. "

[Comment 3] Rather than saying 'various EFs', different subdomains of EFs make more sense to me.

Response: Thank you very much. We changed it to "subdomains of EFs ".

[Comment 4] Need a brief explanation regarding OSE and CSE.

Response: We agree. Accordingly, as this is an essential note for the interpretation of the results, the following sentence has been added to the limitations section: "OSE is in a dynamic environment, where conditions could change any time (i.e., nature or team sports) and CSE are characterized as static with defined and known conditions (i.e., swimming or running). "

[Comment 5] Lines 49-50: Which kind of EFs? Need to clarify specific subdomains of EFs.

Response: Again, we agree; we specified the "subdomains "of EFs in brackets.

[Comment 6] Lines 58-60: This statement needs reference(s).

Response: We agree on that. We added the following references:

REF:

Honeybourne, J. Acquiring skill in sport: An introduction, Repr; Routledge: London [u.a.], 2008, ISBN 9780415349369.

Jacobson, J.; Matthaeus, L. Athletics and executive functioning: How athletic participation and sport type correlate with cognitive performance. Psychology of Sport and Exercise 2014, 15, 521–527, doi:10.1016/j.psychsport.2014.05.005

[Comment 7] Lines 70-78: This paragraph needs reference(s).

Response: Thank you very much for this comment. We think that the reference to Honeybourne fits best here. The continuum of OSE and CSE is explained here in detail. We added the following sentence: "A similar continuum model without the specified categories was introduced by Honeybourne [REF]. "

[Comment 8] Lines 61-78: To me, the mentioning of 'Category' is pretty abrupt, meaning that they were presented without background information. It would be great if authors can add some more information about this.

Response: We think this could be a misunderstanding. In the paragraphs, we explained the categories in lines 65-74 and 75-83 in detail. We give examples for the categories such as "In Category 3 of open-skill sports, athletes can foresee situational conditions to a limited extent only (e.g., in nature sports such as surfing and skiing). "

[Comment 9] Lines 79-80: Since this study is focusing on executive function, executive function can appear from the first, not starting with cognitive function. There are so many subdomains under cognitive function. Starting with executive function will make the readers more focused.

Response: Thank you very much for reading the paper that precisely. We agree on that. We revised the manuscript regarding this comment because we think that at some points, this could be a misunderstanding.

[Comment 10] Lines 87-91: To my understanding, OSE leads to higher cognitive function because it induces greater cognitive demand? Please clarify this in this paragraph.

Response: Yes, you're right. We think the following passage could help the reader understand this part of the text: "OSE could lead to a better performance in EF tasks because the particular sport modes require higher cognitive demands. "

[Comment 11] Lines 131-132: Again, need to clarify categories 1-4.

Response: Thank you very much for the comment. We clarified the categories in lines 65-83 (see comment 5) and added examples for the second comparison between cat. 1 and cat. 4: "We hypothesized that the effect sizes of group differences in studies on sport modes within categories 2 and 3 (e.g., athletics vs. canoe slalom) are smaller when compared to studies on sports within categories 1 and 4 (e.g., swimming vs. basketball). "

[Comment 12] In general, the introduction is quite long and there are too many paragraphs. Authors need to make it more concise with clear background information, gaps in knowledge, aims and hypothesis.

Response: Thank you very much for this comment. We agree with the statement that the introduction is long. However, the first part is necessary for understanding the various sports modes and their classification. The second part shows how this paper stands out from other works.

[Comment 13] Calendar year was used to examine the moderating effects.  This does make sense given the variability in the age range across the studies. However, considering that this study is focused on athletes, their practicing year is as important as their calendar year?

Response: We agree with this and know about this issue. The problem was that only a few studies reported the practicing age. Therefore, it was not possible to implement this as a variable. The level of experience or performance was considered in the qualitative analysis (lines 258-261) and was one selection criterion (see line 163). The definition of athlete was added (line 34). This could be the only affordance for describing the level because there is a lack of detailed reports of the level of performance or competition in most studies. We added this to the first point of limitations of the study (lines 416-418).

[Comment 14] Can training hours/week and IPAQ be used as to determine 'athletes'? The definition of 'athlete' should include both high duration and intensity of training as well as participating in regional/national/international competitions. This definition tool questions me if the studies the authors selected can be analyzed under the same category of athletes.

Response: We highly appreciate this comment. For response, see comment 1 (methods).

[Comment 15] Line 356: I could follow the previous sentences regarding category 1-4, but I could not follow the last sentence. What does it mean by '… overestimated in a narrow sense'?

Response: We apologize for this confusion. We revised the sentence as follows: ". Due to the very different requirements of sports, the effect of CSE vs. OSE could be overestimated when comparing category 1 with category 4. "

[Comment 16] Lines 374-381: For the OSE sports including tennis, table tennis, and badminton, they are more 1 to 1 sports wile basketball and handball are more team sports. Do authors think the individual vs team sports affect the present analysis?

Response: No, we do not think this should lead to differences in various game sports. The requirements are similar, and they are confronted constantly with these demanding situations.

[Comment 17] Line 397-398: The authors just stated that working memory was not studied in an independent manner? Do authors think working memory should be studied separately? If so, what is the rationale?

Response: We apologize for this confusion. We removed the second part of the sentence (line 408). We only meant that it would be better to compare the three commonly described subdomains of EFs (working memory, inhibition, and cognitive flexibility) instead of examining visuospatial attention and processing speed.

Reviewer 2 Report

The review is timely and addresses interesting questions concerning the effects of sport training on fundamental components of EF. The quantitative methodology is first rate and reflects the authors’ expertise. They add to the literature by addressing a long-standing question. Despite the lack of evidence that supports the transfer of mental operations from one skill to a different task, the readership may benefit from the authors’ work. One issue not addressed fully in the Discussion is the role of contextual interference and how sport skills are typically taught. Regardless of the skill category, teachers typically employ blocked methods of training during early stages of practice (favoring closed-skill development). As skill emerges and depending on the goals of the sport, training shifts toward variable methods of training (favoring open-loop skill development). How might the authors interpret the role of contextual interference on during the development of EF? The readership might be interested in the authors’ views on training methods and how they may change over years of practice and how they might differentially influence EF. Consider that the coaches of very elite athletes typically voice the importance of the “fundamentals” of athletic sport skills and the need to practice movements in close-skill conditions.

I hope that the authors continue their line of investigation as it is value to both basic research scientists and applied sport instructors.

Small point: Table 2 is in need of formatting.

Author Response

Comments to Reviewer 2

[General Comment 1] The review is timely and addresses interesting questions concerning the effects of sport training on fundamental components of EF. The quantitative methodology is first rate and reflects the authors' expertise. They add to the literature by addressing a long-standing question. Despite the lack of evidence that supports the transfer of mental operations from one skill to a different task, the readership may benefit from the authors' work.

Response: Thank you very much for this rating.

[General Comment 2] One issue not addressed fully in the Discussion is the role of contextual interference and how sport skills are typically taught. Regardless of the skill category, teachers typically employ blocked methods of training during early stages of practice (favoring closed-skill development). As skill emerges and depending on the goals of the sport, training shifts toward variable methods of training (favoring open-loop skill development). How might the authors interpret the role of contextual interference on during the development of EF? The readership might be interested in the authors' views on training methods and how they may change over years of practice and how they might differentially influence EF. Consider that the coaches of very elite athletes typically voice the importance of the "fundamentals" of athletic sport skills and the need to practice movements in close-skill conditions.

Response: Thank you very much for this general comment and the suggestions. We thought about the type of coaching of the different sports, but we didn't follow this in such a way that we included this consideration in the study. Nevertheless, we included the idea in two places in the manuscript (limitations and conclusions):

"Secondly, the type of practice could not be considered in the recent study. Of course, in OSE, athletes sometimes have to train skills in a closed-skill setting, especially in the first years of practice, but this fact could not be examined in the current analysis. "

"Furthermore, research has to deal with different coaching methods. It would be interesting to compare EFs of athletes practicing in a closed- vs. open-skill development. "

[General Comment 3] I hope that the authors continue their line of investigation as it is value to both basic research scientists and applied sport instructors.

Response: Thank you very much for these kind words.

[Comment 1] Small point: Table 2 is in need of formatting.

Response: We formatted Table 2.

Round 2

Reviewer 1 Report

Introduction

1.       Line 34: I am not sure if I agree with the author’s definition of athletes. My previous comment 14 persists. Only proficiency in sports justifies being athletes?

2.       Lines 92-94: Need to add reference(s) for this statement. Also, which kind of cognitive demands the authors are referring to? Need to elaborate.

Discussion

1.       Line 365: What does it mean by ‘Due to the very different requirements…’?

2.       Lines 417-421: I think these sentences are not clear. Please consider revising these.

Author Response

Comments to Reviewer 1

[Comment 1] Line 34: I am not sure if I agree with the author’s definition of athletes. My previous comment 14 persists. Only proficiency in sports justifies being athletes?

Response: Thank you very much for this comment. We appreciate the work and think we could strengthen this statement by implementing the established definition of Araújo and Scharhag [REF].

REF: Araújo, C.G.S.; Scharhag, J. Athlete: a working definition for medical and health sciences research. Scand. J. Med. Sci. Sports 2016, 26, 4–7, doi:10.1111/sms.12632.

[Comment 2] Lines 92-94: Need to add reference(s) for this statement. Also, which kind of cognitive demands the authors are referring to? Need to elaborate.

Response: We agree with this statement. We cited a reference [8] where we explained the relation between the requirements in the particular sport and EFs. We hope we can make this statement clearer for the reader. It was cited in the manuscript before.

The part was revised as follows: “OSE could lead to a better performance in EF tasks because the particular sport modes re-quire higher cognitive demands, especially with regard to EFs. For example, if a soccer player plans a pass to a teammate and an opponent covers the passing line, the player must inhibit his initially planned movement (inhibition) [9].”

9. Heilmann, F.; Wollny, R.; Lautenbach, F. Inhibition and Calendar Age Explain Variance in Game Performance of Youth Soccer Athletes. Int. J. Environ. Res. Public Health 2022, 19, doi:10.3390/ijerph19031138.

[Comment 3] Line 365: What does it mean by ‘Due to the very different requirements…’?

Response: Thank you again for reading the manuscript so carefully. We agree that this sentence is a bit confusing. Therefore, we revised it as follows:

“Due to the different cognitive requirements in these categories, the effect of CSE vs. OSE could be overestimated when comparing category 1 with category 4. However, the effect should be much higher than when comparing categories 2 and 3.”

[Comment 4] Lines 417-421: I think these sentences are not clear. Please consider revising these.

Response: Thank you very much. We think that the structure of the mentioned sentences was a bit confusing, and we revised them as follows:

“First, only the cognitive demands but not the physical fitness (e.g., aerobic or muscular fitness) or level of performance or competition, as well as levels of endocrine hormone status reported in the studies, were considered when examining the effects of OSE and CSE on EFs. Due to the heterogeneous types of measurement and the description of the mentioned factors cannot be included in the calculation.”